# Making Translators Privacy-aware on the User's Side

## Abstract

We propose PRISM to enable users of machine translation systems to preserve the privacy of data on their own initiative. There is a growing demand to apply machine translation systems to data that require privacy protection. While several machine translation engines claim to prioritize privacy, the extent and specifics of such protection are largely ambiguous. First, there is often a lack of clarity on how and to what degree the data is protected. Even if service providers believe they have sufficient safeguards in place, sophisticated adversaries might still extract sensitive information. Second, vulnerabilities may exist outside of these protective measures, such as within communication channels, potentially leading to data leakage. As a result, users are hesitant to utilize machine translation engines for data demanding high levels of privacy protection, thereby missing out on their benefits. PRISM resolves this problem. Instead of relying on the translation service to keep data safe, PRISM provides the means to protect data on the user's side. This approach ensures that even machine translation engines with inadequate privacy measures can be used securely. For platforms already equipped with privacy safeguards, PRISM acts as an additional protection layer, reinforcing their security furthermore. PRISM adds these privacy features without significantly compromising translation accuracy. Our experiments demonstrate the effectiveness of PRISM using real-world translators, T5 and ChatGPT (GPT-3.5-turbo), and the datasets with two languages. PRISM effectively balances privacy protection with translation accuracy over other user-side privacy protection protocols and helps users grasp the content written in a foreign language without leaking the original content.

## 1 Introduction

Machine translation systems are now essential in sectors including business and government for translating materials such as e-mails and documents [8, 41, 42]. Their rise in popularity can be attributed to recent advancements in language models [4, 31, 40] that have significantly improved translation accuracy, enhancing their overall utility. There is a growing demand to use these tools for private and sensitive information. For instance, office workers often need to translate e-mails from clients in other countries, but they want to keep these e-mails secret. Many are worried about using machine translation because there's a chance the information might get leaked. This means that, even with these helpful tools around, people often end up translating documents by themselves to keep the information safe.

Although many machine translation platforms claim they value privacy, the details and depth of this protection are not always clear. First, it's often uncertain how and to what level the data is kept safe. The details of the system are often an industrial secret of the service provider, and the source code is rarely disclosed. Even if providers are confident in their security, sophisticated attackers might still access private information. Also, there could be risks outside of these safeguards, including during data transfer, leading to potential leaks. Because of these concerns, users are cautious about using translation tools for sensitive data, missing out on their benefits.

In response to the prevalent concerns regarding data security in machine translation, we present PRISM (PRIvacy Self Management), which empowers users to actively manage and ensure the protection of their data. Instead of placing complete trust in the inherent security protocols of translation platforms, PRISM provides users with mechanisms for personal data safeguarding. This proactive strategy allows users to

confidently use even translation engines that may not offer privacy measures. For platforms already equipped with privacy safeguards, PRISM acts as an additional protection layer, reinforcing their security mechanisms. PRISM adds these privacy features without much degradation of translation accuracy.

We propose two variants of PRISM. PRISM-R is a simple method with a theoretical guarantee of differential privacy. PRISM* (PRISM-Star) is a more sophisticated method that can achieve better translation accuracy than PRISM-R at the price of losing the theoretical guarantee. In practice, we recommend using PRISM* for most use cases and PRISM-R for cases where the theoretical guarantee is required.

In the experiments, we use real-world translators, namely T5 [32] and ChatGPT (GPT-3.5-turbo) [18, 30], and the English $\rightarrow$ French and English $\rightarrow$ German translation. We confirm that PRISM can effectively balance privacy protection with translation accuracy over other user-side privacy protection protocols. PRISM helps users grasp the content written in a foreign language without leaking the original content.

The contributions of this paper are as follows:

- We formulate the problem of user-side realization of data privacy for machine translation systems.

- We propose PRISM, which enables users to preserve the privacy of data on their own initiative.

- We formally show that PRISM can preserve the privacy of data in terms of differential privacy.

- We propose an evaluation protocol for user-side privacy protection for machine translation systems.

- We confirm that PRISM can effectively balance privacy protection with translation accuracy using the real-world ChatGPT translator.

**Reproducibility**: Our code and trained dictionaries are available at `https://github.com/xxxxxx/prism` (to be filled in the camera-ready).

## 2 Problem Formulation

We assume that we have access to a black-box machine translation system $T$ that takes a source text $x$ and outputs a target text $y$. In practice, $T$ can be ChatGPT[30], DeepL[10], or Google Translate[16]. We assume that the quality of the translation $T(x)$ is satisfactory, but $T$ may leak information or be unreliable in terms of privacy protection. Therefore, it is crucial to avoid feeding sensitive text $x$ directly into $T$. We have a sensitive source text $x_{\mathrm{pri}}$, and our goal is to safely translate $x_{\mathrm{pri}}$. We also assume that we have a dataset of non-sensitive source texts $\mathcal{D} = \{x_1, \ldots, x_n\}$. $\mathcal{D}$ is unlabeled and need not be relevant to $x_{\mathrm{pri}}$. Therefore, it is cheap to collect $\mathcal{D}$. In practice, $\mathcal{D}$ can be public news texts, and $x_{\mathrm{pri}}$ can be an e-mail.

When considering user-side realization, the method should be simple enough to be executed on the user's side. For example, it is difficult for users to run a large language model or to train a machine learning model on their own because it requires a lot of computing resources and advanced programming skills. Therefore, we stick to simple and accessible methods.

In summary, our goal is to safely translate $x_{\mathrm{pri}}$ using $T$ and $\mathcal{D}$, and the desiderata of the method are summarized as follows:

**Accurate** The final output should be a good translation of the input text $x_{\mathrm{pri}}$.

**Secure** The information passed to $T$ should not contain much information of the input text $x_{\mathrm{pri}}$.

**Simple** The method should be lightweight enough for end-users to use.

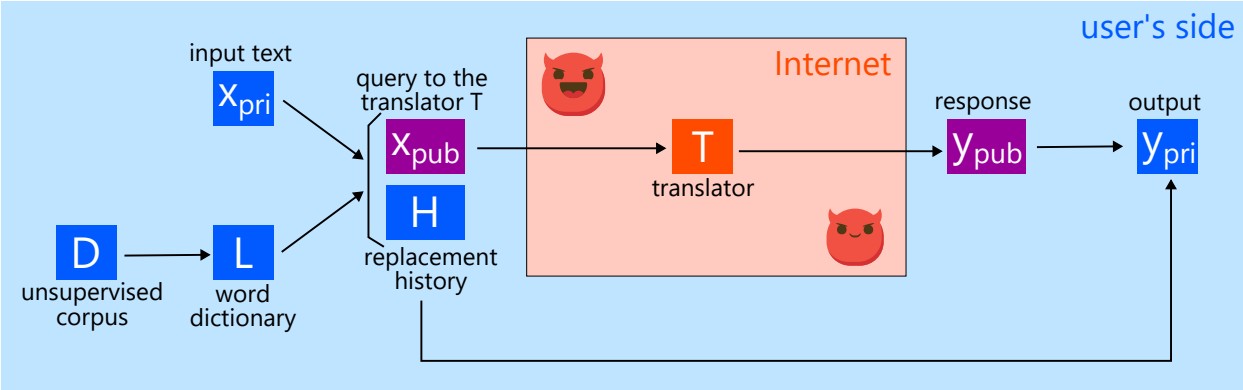

Figure 1: Overview of PRISM. The blue boxes indicate information kept on the user's side, the purple boxes indicate information exposed to the Internet, and the red region indicates the Internet. The purple boxes should not contain much information about the input text $x_{\text{pri}}$.

## 3 Proposed Method (PRISM)

### 3.1 Overview

PRISM has four steps as shown in Figure 1. (i) PRISM creates a word translation dictionary using $T$ and $\mathcal{D}$. This step should be done only once, and the dictionary can be used for other texts and users. (ii) PRISM converts the source text $x_{\text{pri}}$ to a non-sensitive text $x_{\text{pub}}$. (iii) PRISM translates $x_{\text{pub}}$ to $y_{\text{pub}}$ using $T$. (iv) PRISM converts $y_{\text{pub}}$ to $y_{\text{pri}}$ using the replacement history $\mathcal{H}$. We explain each step in detail in the following.

Let us first illustrate the behavior of PRISM with an example. let $x_{\text{pri}}$ be "Alice is heading to the hideout." and $T$ be a machine translation system from English to French. PRISM converts $x_{\text{pri}}$ to $x_{\text{pub}} =$ "Bob is heading to the store," which is not sensitive and can be translated with $T$. PRISM temporarily stores the substitutions (Alice → Bob) and (base → restaurant). Note that this substitution information is kept on the user's side and is not passed to $T$. Then, PRISM translates $x_{\text{pub}}$ to $y_{\text{pub}} =$ "Bob se dirige vers la boutique." using the translator $T$. Finally, PRISM converts $y_{\text{pub}}$ to $y_{\text{pri}} =$ "Alice se dirige vers la cachette." using the word translation dictionary, Alice (En) → Alice (Fr), Bob (En) → Bob (Fr), store (En) → boutique (Fr), and hideout (En) → cachette (Fr). The final output $y_{\text{pri}}$ is the translation of $x_{\text{pri}}$, and PRISM did not pass the information that Alice is heading to the hideout to $T$.

### 3.2 Word Translation Dictionary

We assume that a user does not have a word translation dictionary for the target language. We propose to create a word translation dictionary using the unsupervised text dataset $\mathcal{D}$. The desideratum is that the dictionary should be robust. Some words have multiple meanings, and we want to avoid incorrect substitutions in PRISM. Let $\mathcal{V}$ be the vocabulary of the source language. Let $S$ be a random variable that takes a random sentence from $\mathcal{D}$, and let $S_w$ be the result of replacing a random word in $S$ with $w \in \mathcal{V}$. We translate $S$ to the target language and obtain $R$ and translate $S_w$ to obtain $R_w$. Let

$$p_{w,v} \overset{\text{def}}{=} \frac{\Pr[v \in R_w]}{\Pr[v \in R]} \tag{1}$$

be the ratio of the probability of $v$ appearing in $R_w$ to the probability of $v$ appearing in $R$. The higher $p_{w,v}$ is, the more likely $v$ is the correct translation of $w$ since $v$ appears in the translation if and only if $w$ appears in the source sentence. Note that if we used only the numerator, article words such as "la" and "le" would have high scores, and therefore we use the ratio instead. Let $L(w)$ be the list of words in the decreasing order of $p_{w,v}$. $L(w, 1)$ is the most likely translation of $w$, and $L(w, 2)$ is the second most likely translation of $w$, and so on.

---

**Algorithm 1:** PRISM-R

---

**Input:** Source text $x_{\text{pri}}$; Word translation dictionary $L$; Ratio $r \in (0, 1)$.
**Output:** Translated text $y_{\text{pri}}$.

**1** $t_1, \ldots, t_n \leftarrow \text{Tokenize}(x_{\text{pri}})$                                 `// Tokenize` $x_{\text{pri}}$
**2** $\mathcal{H} \leftarrow \emptyset$                              `// The history of substitutions`
**3** **for** $i \leftarrow 1$ **to** $n$ **do**
**4**     $p \sim \text{Unif}(0, 1)$
**5**     **if** $p < r$ **then**
**6**         $u_i \leftarrow$ a random source word in $L$            `// Choose a substitution word`
**7**         $\mathcal{H} \leftarrow \mathcal{H} \cup \{(t_i, u_i)\}$                `// Update the history`
**8**         $t_i \leftarrow u_i$
**9** $x_{\text{pub}} \leftarrow \text{Detokenize}(t_1, \ldots, t_n)$                  `// Detokenize` $t_1, \ldots, t_n$
**10** $y_{\text{pub}} \leftarrow T(x_{\text{pub}})$                         `// Translate` $x_{\text{pub}}$
**11** $y_{\text{pri}} \leftarrow y_{\text{pub}}$                             `// Copy` $y_{\text{pub}}$
**12** **for** $(w, u) \in \mathcal{H}$ **do**
**13**     **for** $v \in L(u)$ **do**
**14**         **if** $v \in y_{pri}$ **then**
**15**             $y_{\text{pri}} \leftarrow$ replace $v$ with $L(w, 1)$ in $y_{\text{pri}}$ **break**
**16** **return** $y_{pri}$

---

It should be noted that the translation engine used here is not necessarily the same as the one $T$ we use in the test phase. As we need to translate many texts here, we can use a cheaper translation engine. We also note that once we create the word translation dictionary, we can use it for other texts and users. We will distribute the word translation dictionaries for English $\rightarrow$ French and English $\rightarrow$ German, and users can skip this step if they use these dictionaries.

### 3.3 PRISM-R

PRISM-R is a simple method to protect data privacy on the user's side. PRISM-R randomly selects words $w_1, \ldots, w_k$ in the source text $x_{\text{pri}}$ and randomly selects substitution words $u_1, \ldots, u_k$ from the word translation dictionary. $x_{\text{pub}}$ is the result of replacing $w_1$ with $u_1$, $\ldots$, and $w_k$ with $u_k$. PRISM-R then translates $x_{\text{pub}}$ to $y_{\text{pub}}$ using $T$. Finally, PRISM-R converts $y_{\text{pub}}$ to $y_{\text{pri}}$ as follows. Possible translation words of $u_i$ are $L(u_i)$. PRISM-R first searches for $L(u_i, 1)$, the most likely translation of $u_i$, in $y_{\text{pub}}$. There should be $L(u_i, 1)$ in $y_{\text{pub}}$ if $T$ translated $u_i$ to $L(u_i, 1)$. If $L(u_i, 1)$ is found, PRISM-R replaces $L(u_i, 1)$ with $L(w_i, 1)$. However, if $u_i$ has many translation candidates, $T$ may not have translated $u_i$ to $L(u_i, 1)$. If $L(u_i, 1)$ is not found, it proceeds to $L(u_i, 2)$, the second most likely translation of $u_i$, and replaces $L(u_i, 2)$ with $L(w_i, 1)$, and so on.

The pseudo code is shown in Algorithm 1.

### 3.4 Differential Privacy of PRISM-R

Differential privacy [11] provides a formal guarantee of data privacy. We show that PRISM-R satisfies differential privacy. This result not only provides a privacy guarantee but also shows PRISM-R can be combined with other mechanisms due to the inherent composability and post-processing resilience of differential privacy [24].

We first define differential privacy. We say texts $x = w_1, \ldots, w_n$ and $x' = w'_1, \ldots, w'_n$ are neighbors if $w_i = w'_i$ for all $i \in \{1, \ldots, n\}$ except for one $i \in \{1, \ldots, n\}$. Let $x \sim x'$ denote that $x$ and $x'$ are neighbors. Let $A$ be a randomized mechanism that takes a text $x$ and outputs a text $y$. Differential privacy is defined as follows.

**Definition 3.1** (Differential Privacy). *A satisfies $\epsilon$-differential privacy if for all $x \sim x'$ and $S \subseteq \text{Im}(A)$,*

$$\frac{\Pr[A(x) \in S]}{\Pr[A(x') \in S]} \le e^\epsilon. \tag{2}$$

We show that the encoder of PRISM-R $A_{\text{PRISM-R}} \colon x_{\text{pri}} \mapsto x_{\text{pub}}$ is differential private, and therefore, $x_{\text{pri}}$ cannot be inferred from $x_{\text{pub}}$, which is the only information that $T$ can access.

**Theorem 3.2.** $A_{\text{PRISM-R}}$ *is* $\ln\left(\frac{r+|\mathcal{V}|(1-r)}{r}\right)$-*differential private.*

We emphasize that the additive constant $\delta$ is zero, i.e., PRISM-R is $(\epsilon, 0)$-differential private, which provides a strong guarantee of data privacy.

*Proof.* Let $x = w_1, \dots, w_n$ and $x' = w'_1, \dots, w'_n$ be any two neighboring texts. Without loss of generality, we assume that $w_1 = w'_1, \dots, w_{n-1} = w'_{n-1}$ and $w_n \ne w'_n$. Let $s = s_1, \dots, s_n$ be any text, and let

$$c \overset{\text{def}}{=} \sum_{i=1}^{n} \mathbb{1}[s_i \ne w_i] \tag{3}$$

be the number of different words in $x$ and $s$. The probability that PRISM-R convert $x$ to $s$ is

$$\Pr[A_{\text{PRISM-R}}(x) = s] = \sum_{i=c}^{n} \binom{n-c}{i-c} r^i (1-r)^{n-i} \left(\frac{1}{|\mathcal{V}|}\right)^i, \tag{4}$$

where $i$ is the number of replaced words because all of the different words must be replaced, and the number of ways to choose the remaining words is $\binom{n-c}{i-c}$. This probability can be simplified as follows:

$$\Pr[A_{\text{PRISM-R}}(x) = s] = \sum_{i=c}^{n} \binom{n-c}{i-c} r^i (1-r)^{n-i} \left(\frac{1}{|\mathcal{V}|}\right)^i \tag{5}$$

$$= \sum_{i=0}^{n-c} \binom{n-c}{i} r^{i+c} (1-r)^{n-c-i} \left(\frac{1}{|\mathcal{V}|}\right)^{i+c} \tag{6}$$

$$= r^c (1-r)^{n-c} \left(\frac{1}{|\mathcal{V}|}\right)^c \sum_{i=0}^{n-c} \binom{n-c}{i} \left(\frac{r}{|\mathcal{V}|(1-r)}\right)^i \tag{7}$$

$$= r^c (1-r)^{n-c} \left(\frac{1}{|\mathcal{V}|}\right)^c \left(1 + \frac{r}{|\mathcal{V}|(1-r)}\right)^{n-c}, \tag{8}$$

where we used the binomial theorem in the last equality. Similarly, let

$$c' \overset{\text{def}}{=} \sum_{i=1}^{n} \mathbb{1}[s_i \ne w'_i] \tag{9}$$

$$= \begin{cases} c & \text{if } \mathbb{1}[s_n \ne w_n] = \mathbb{1}[s_n \ne w'_n] = 1 \\ c+1 & \text{if } \mathbb{1}[s_n \ne w_n] = 0 \text{ and } \mathbb{1}[s_n \ne w'_n] = 1 \\ c-1 & \text{if } \mathbb{1}[s_n \ne w_n] = 1 \text{ and } \mathbb{1}[s_n \ne w'_n] = 0 \end{cases}. \tag{10}$$

be the number of different words in $x'$ and $s$. Then,

$$\Pr[A_{\text{PRISM-R}}(x') = s] = r^{c'} (1-r)^{n-c'} \left(\frac{1}{|\mathcal{V}|}\right)^{c'} \left(1 + \frac{r}{|\mathcal{V}|(1-r)}\right)^{n-c'}. \tag{11}$$

---

**Algorithm 2:** PRISM*

**Input:** Source text $x_{\text{pri}}$; Word translation dictionary $L$; Confidence Scores $c$, Ratio $r \in (0,1)$.
**Output:** Translated text $y_{\text{pri}}$.

**1** $t_1, \ldots, t_n \leftarrow \text{Tokenize}(x_{\text{pri}})$                  `// Tokenize `$x_{\text{pri}}$
**2** $s_1, \ldots, s_n \leftarrow \text{Part-of-Speech}(t_1, \ldots, t_n)$
**3** $k \leftarrow 0$            `// The number of substitutions`
**4** $\mathcal{H} \leftarrow \emptyset$            `// The history of substitutions`
**5** **for** $i \in \{1, \ldots, n\}$ in the decreasing order of $c(t_i, s_i)$ **do**
**6**      $u_i \leftarrow$ the unused source word $u_i$ with the highest confidence score $c(u_i, s_i)$ in $L$      `// Choose a`
         `substitution word`
**7**      $\mathcal{H} \leftarrow \mathcal{H} \cup \{(t_i, u_i, s_i)\}$            `// Update the history`
**8**      $t_i \leftarrow u_i$
**9**      $k \leftarrow k + 1$
**10**      **if** $k \geq rn$ **then**
**11**          **break**

**12** $x_{\text{pub}} \leftarrow \text{Detokenize}(t_1, \ldots, t_n)$            `// Detokenize `$t_1, \ldots, t_n$
**13** $y_{\text{pub}} \leftarrow T(x_{\text{pub}})$            `// Translate `$x_{\text{pub}}$
**14** $y_{\text{pri}} \leftarrow y_{\text{pub}}$            `// Copy `$y_{\text{pub}}$
**15** **for** $(w, u, s) \in \mathcal{H}$ **do**
**16**      **for** $v \in L(u, s)$ **do**
**17**          **if** $v \in y_{pri}$ **then**
**18**              $y_{\text{pri}} \leftarrow$ replace $v$ with $L(w, s, 1)$ in $y_{\text{pri}}$ **break**

**19** **return** $y_{pri}$

---

Combining Eqs. (8) and (11),

$$\frac{\Pr[A_{\text{PRISM-R}}(x) = s]}{\Pr[A_{\text{PRISM-R}}(x') = s]} = \begin{cases} 1 & \text{if } \mathbb{1}[s_n \neq w_n] = \mathbb{1}[s_n \neq w'_n] = 1 \\ \frac{r + |\mathcal{V}|(1-r)}{r} & \text{if } \mathbb{1}[s_n \neq w_n] = 0 \text{ and } \mathbb{1}[s_n \neq w'_n] = 1 \\ \frac{r}{r + |\mathcal{V}|(1-r)} & \text{if } \mathbb{1}[s_n \neq w_n] = 1 \text{ and } \mathbb{1}[s_n \neq w'_n] = 0 \end{cases} \tag{12}$$

$$\leq \frac{r + |\mathcal{V}|(1-r)}{r}. \tag{13}$$

$\square$

An interesting part of PRISM is that PRISM is resilient against the purturbation due to the final substitution step. Many of differential private algorithms add purturbation to the data [1, 3, 7, 13, 43] and therefore, their final output becomes unreliable when the privacy constraint is severe. By contrast, PRISM enjoys both of the privacy guarantee and the reliability of the final output thanks to the purturbation step and the recovery step. The information $x_{\text{pub}}$ passed to $T$ has little information due ot the purturbation step. This, however, makes the intermediate result $y_{\text{pub}}$ an unreliable translation of $x_{\text{pri}}$. PRISM recovers a good translation $y_{\text{pri}}$ by the final substitution step.

### 3.5 PRISM*

PRISM* is a more sophisticated method and achieves better accuracy than PRISM-R. PRISM* chooses words $w_1, \ldots, w_k$ in the source text $x_{\text{pri}}$ and substitution words $u_1, \ldots, u_k$ from the word translation dictionary more carefully while PRISM-R chooses them randomly to achieve differential privacy. PRISM* has two mechanisms to choose words. The first mechanism is to choose words so that the part-of-speech tags match. The second mechanism is to choose words that can be translated accurately by the word dictionary. We explain each mechanism in detail in the following.

PRISM* creates a word translation dictionary with a part of speech tag. The procedure is the same as Section 3.2 except that we use the part-of-speech tag of the source word as the key of the dictionary. Let $(w, s)$ be a pair of a source word $w$ and its part-of-speech tag $s$. PRISM* replaces a random word with part-of-speech tag $s$ with $w$ to create $S_{w,s}$, obtains $R_{w,s}$ by translating $S_{w,s}$, and defines

$$p_{w,s,v} \stackrel{\text{def}}{=} \frac{\Pr[v \in R_{w,s}]}{\Pr[v \in R]}. \tag{14}$$

$L(w, s)$ is the list of words $v$ in the decreasing order of $p_{w,s,v}$.

In the test time, PRISM* chooses substitute words so that the part-of-speech tags match, and use the word translation dictionary with part-of-speech tags to determine the translated word.

PRISM* also uses the confidence score

$$c(w, s) \stackrel{\text{def}}{=} \max_v p_{w,s,v}, \tag{15}$$

which indicates the reliability of word translation $w \to L(w, 1)$, to choose words. Multiple-meaning words should not be substituted in PRISM because a word-to-word translation may fail. PRISM* chooses words to be substituted in the decreasing order of the confidence score, which results in selecting single-meaning and reliable words that can be translated accurately by the word dictionary. If a word $(w, s)$ has two possible translations $v_1$ and $v_2$ that are equally likely, $\Pr[v \in R_{w,s}]$ is lower than 0.5 for any $v$, even for $v = v_1$ and $v = v_2$, the confidence score $c(w, s)$ tends to be low, and PRISM* avoids selecting such $(w, s)$. The selected words $w$ can be reliably translated by $L(w, s, 1)$. PRISM* also chooses substitute words with high confidence scores so that PRISM* can robustly find the corresponding word $L(w, s, 1)$ in the translated text $y_{\text{pub}}$ in the final substitution step.

The pseudo code of PRISM* is shown in Algorithm 2.

Note that PRISM* does not enjoy the differential privacy guarantee of PRISM-R as (i) PRISM* replaces words with the same part-of-speech tag so that two texts with different part-of-speech templates have zero probability of transition, and (ii) PRISM* chooses words with high confidence scores so that the probability of transition is biased. Nevertheless, PRISM* empirically strikes a better trade-off between privacy and accuracy than PRISM-R as we will show in the experiments. Note that PRISM* can be combined with PRISM-R to guarantee differential privacy. For example, one can apply PRISM-R and PRISM* in a nested manner, which guarantees differential privacy due to the differential privacy of PRISM-R (Theorem 3.2) and the post processing resilience of differential privacy [24]. One can also apply PRISM-R with probability $(1 - \beta)$ and PRISM* with probability $\beta$, which also guarantees differential privacy because the minimum probability of transition is bounded from below due to the PRISM-R component.

## 4 Experiments

We confirm the effectiveness of our proposed methods through experiments.

### 4.1 Evaluation Protocol

As our problem setting is novel, we first propose an evaluation protocol for the user-side realization of privacy-aware machine translation systems. We evaluate the translation accuracy and privacy protection as follows.

Let $\mathcal{X} = \{x_1, x_2, \ldots, x_N\}$ be a set of test documents to be translated. Our aim is to read $\mathcal{X}$ in the target language without leaking information of $\mathcal{X}$.

For evaluation purposes, we introduce a question-answering (QA) dataset $\mathcal{Q} = \{(q_{ij}, a_{ij})\}$, where $q_{ij}$ and $a_{ij}$ are a multiple-choice question and answer regarding the document $x_i$, respectively. $\mathcal{Q}$ is shown only to the evaluator, and not to the translation algorithm.

**Privacy-preserving Score.** The idea of our privacy score is based on an adversarial evaluation where adversaries try to extract information from the query sent by the user. Let $x_i^{\text{pub}}$ be the query sent to

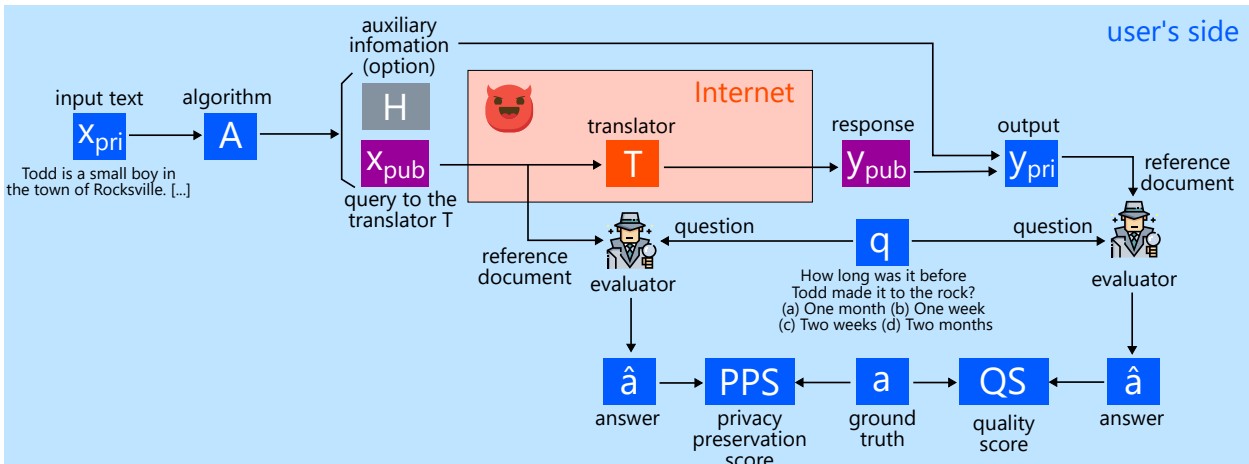

Figure 2: Overview of the evaluation protocol.

the translator $T$. An evaluator is given $x_i^{\text{pub}}$ and $q_{ij}$, and asked to answer the question. The privacy-preserving score of the translation algorithm is defined as $\text{PPS} = (1 - \text{acc})$, where acc is the accuracy of the evaluator. The higher the privacy-preserving score is, the better the privacy protection is. Intuitively, if the accuracy is low, the evaluator cannot draw any information from $x_i^{\text{pub}}$ to answer the question. Conversely, if the accuracy is high, the evaluator can infer the answer solely from $x_i^{\text{pub}}$, which means that $x_i^{\text{pub}}$ leaks information. We note that the translation algorithm does not know the question $q_{ij}$, and therefore, the translation algorithm needs to protect all information to achieve a high privacy-preserving score so that any answer on the document cannot be drawn from $x_i^{\text{pub}}$. The rationale behind this score is that we cannot predict what form information leaks will take in advance. Even if $x_{\text{pub}}$ does not look like $x_{\text{pri}}$ at a glance, sophisticated adversaries might extract information that can be used to infer $x_{\text{pri}}$. Therefore, we employ an outside evaluator and adopt an adversarial evaluation.

**Quality Score.** Let $y_i^{\text{pri}}$ be the final output of the translation algorithm. We use the same QA dataset and ask an evaluator to answer the question $q_{ij}$ using $y_i^{\text{pri}}$. The quality score of the translation algorithm is defined as $\text{QS} = \text{acc}$, where acc is the accuracy of the evaluator. The higher the quality score is, the better the translation quality is. Intuitively, if the accuracy is high, the evaluator can answer the question correctly using $y_i^{\text{pri}}$, which means that $y_i^{\text{pri}}$ contains sufficient information on $x_i$. We note again that the translation algorithm does not know the question $q_{ij}$, and therefore, the translation algorithm needs to preserve all information to achieve a high quality score so that any answer on the document can be drawn from $y_i^{\text{pri}}$.

The protocol is illustrated in Figure 2.

We introduce the area-under-privacy-quality curve (AUPQC) to measure the effectiveness of methods. The privacy-preserving score and the quality score are in a trade-off relationship. Most methods, including PRISM-R and PRISM*, have a parameter to control the trade-off. An effective method should have a high privacy-preserving score and a high quality score at the same time. We use the AUPQC to measure the trade-off. Specifically, we scan the trade-off parameter and plot the privacy-preserving score and the quality score in the two-dimensional space. The AUPQC is the area under the curve. The larger the AUPQC is, the better the method is. The pseudo code is shown in Algorithm 3.

We also introduce QS@p, a metric indicating the quality score at a specific privacy-preserving score. The higher QS@p is, the better the method is. In realistic scenarios, we may have a severe security budget $p$ which represents the threshold of information leakage we can tolerate. QS@p is particularly useful under such constraints as it provides a direct measure of the quality we can enjoy under the security budget. It is noteworthy that the privacy-preserving score can be evaluated before we send information to the translator $T$. Therefore, we can tune the trade-off parameter and ensure that we enjoy the privacy-preserving score = $p$ and the quality score = QS@p.

---

**Algorithm 3:** AUPQC

---

**1** $\mathcal{P} \leftarrow$ the set of the trade-off parameters in the increasing order of the privacy-preserving score.

**2** $s \leftarrow 0$                                   `// The area under the curve`

**3** $(\text{PPS}_{\text{prev}}, \text{QS}_{\text{prev}}) \leftarrow (\text{None}, \text{None})$

**4 for** $\alpha \leftarrow \mathcal{P}$ **do**

**5**      $(\text{PPS}, \text{QS}) \leftarrow \text{Evaluate}(\alpha)$      `// Evaluate the privacy-preserving score and the quality score`

**6**      **if** $PPS_{prev}$ is None **then**

**7**          $s \leftarrow s + \text{PPS} \times \text{QS}$                          `// Add the area of the first rectangle`

**8**      **else**

**9**          $s \leftarrow s + (\text{PPS} - \text{PPS}_{\text{prev}}) \times (\text{QS}_{\text{prev}} + \text{QS})/2$      `// Add the area of the trapezoid`

**10**      $(\text{PPS}_{\text{prev}}, \text{QS}_{\text{prev}}) \leftarrow (\text{PPS}, \text{QS})$

**11 return** $s$

---

## 4.2 Experimental Setups

We use the MCTest dataset [33] for the documents $x_i$, question $q_{ij}$, and answer $a_{ij}$. Each document in the MCTest dataset is a short story with four questions and answers. The reason behind this choice is that the documents of the MCTest dataset were original ones created by crowdworkers. This is in contrast to other reading comprehension datasets such as NarrativeQA [19] and CBT [17] datasets, which are based on existing books and stories, where the evaluator can infer the answers without relying on the input document $x_i^{\text{pub}}$.

We use T5 [32] and GPT-3.5-turbo [30] as the translation algorithm $T$. We use the prompt "Directly translate English to [Language]: [Source Text]" to use GPT-3.5-turbo for translation.

We also use GPT-3.5-turbo as the evaluator. Specifically, the prompt is composed of four parts. The first part of the prompt is the instruction "Read the following message and solve the following four questions." The second part is the document to be evaluated, which is the query document $x_i^{\text{pub}}$ for the privacy-preserving score and the final output $y_i^{\text{pri}}$ for the quality score. The third part is the four questions. The last part is the instruction "Output only four characters representing the answers, e.g.,\n1. A\n2. B\n3. A\n4. D." We parse the output of GPT-3.5-turbo to extract the answers and evaluate the accuracy.

We use the following four methods.

**Privacy- and Utility-Preserving Textual Analysis (PUP)** [13] is a differential private algorithm to convert a document to a non-sensitive document without changing the meaning of $x$. PUP has a trade-off parameter $\lambda$ for privacy and utility. We convert the source text $x_{\text{pri}}$ to $x_{\text{pub}}$ using PUP and translate $x_{\text{pub}}$ to obtain the final output $y_{\text{pri}}$.

**NoDecode** translates the encoded text $x_{\text{pub}}$ of PRISM* to obtain the final output $y_{\text{pri}}$. NoDecode does not decode the output of the translator $T$. This method has the same privacy-preserving property as PRISM* but the accuracy should be lower. The improvements from NoDecode are the contribution of our framework.

**PRISM-R** is our method proposed in Section 3.3.

**PRISM\*** is our method proposed in Section 3.5.

We change the ratio $r$ of NoDecode, PRISM-R, and PRISM* and the parameter $\lambda$ of PUP to control the trade-off between privacy-preserving score and the quality score.

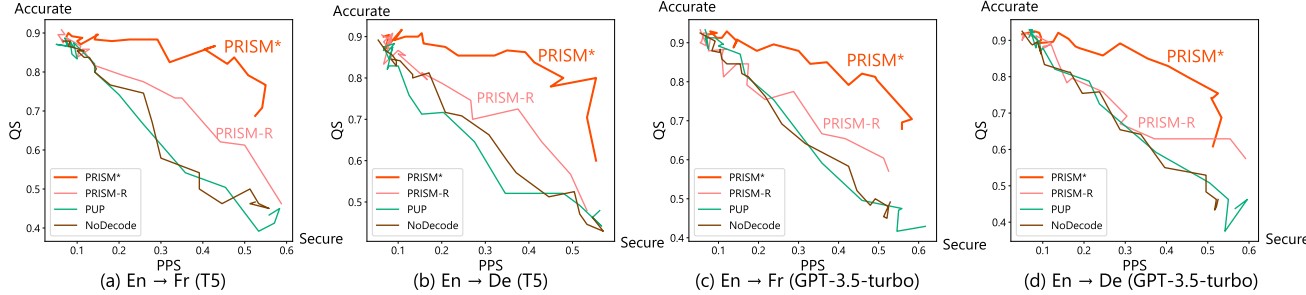

Figure 3: Trade-off between the privacy-preserving score and the quality score. The x-axis is the privacy-preserving score and the y-axis is the quality score.

Table 1: Quantitative Results. The best results are shown in **bold**, and the second best results are shown in underline.

|  | En → Fr (T5) | | En → De (T5) | | En → Fr (ChatGPT) | | En → De (ChatGPT) | |
|---|---|---|---|---|---|---|---|---|
|  | AUPQC ↑ | QS@0.5 ↑ | AUPQC ↑ | QS@0.5 ↑ | AUPQC ↑ | QS@0.5 ↑ | AUPQC ↑ | QS@0.5 ↑ |
| NoDecode | 0.355 | 0.493 | 0.373 | 0.524 | 0.376 | 0.495 | 0.370 | 0.480 |
| PUP | 0.363 | 0.439 | 0.363 | 0.505 | 0.415 | 0.487 | 0.391 | 0.511 |
| PRISM-R | 0.431 | 0.613 | 0.396 | 0.557 | 0.399 | 0.611 | 0.432 | 0.629 |
| PRISM* | **0.454** | **0.803** | **0.473** | **0.789** | **0.482** | **0.799** | **0.445** | **0.769** |

### 4.3 Results

Figure 3 shows the trade-off, where the x-axis is the privacy-preserving score and the y-axis is the quality score. PRISM* clearly strikes the best trade-off, and the results of PRISM-R are also better than those of NoDecode and PUP, especially when the privacy-preserving score is high.

The maximum privacy-preserving score is around 0.5 for all methods, even though there are four choices in each question. Intuitively, the accuracy of the evaluator should be 0.25 when the reference document is random, so the maximum PPR should be 0.75. We found that this is because some questions can be inferred solely from the question text. For example, there is a question "How did the girl hurt her knee? (a) she was in the street (b) she had no friends (c) she fell down, and (d) the old lady's bike hit her." We can infer the answer is (c) or (d) as (a) and (b) do not make sense (the answer is (c)). To verify this hypothesis, we had GPT-3.5-turbo answer the questions using only the question text. The accuracy was 0.492. Therefore, PPS ≈ 0.5 indicates that the query has no more information than the empty text. This experiment also shows that the GPT-3.5-turbo evaluator is so powerful that it can infer the answer from the question text only, and it is an effective adversarial evaluator.

Table 1 shows the quantitative results. We report QS@0.5, i.e., the quality score when PPS is 0.5, which roughly means the quality we can enjoy when no information is leaked based on the above analysis. PRISM* consistently achieves the best scores across all the metrics and settings, and PRISM-R achieves the second-best results in most of the metrics and settings. Notably, PRISM* achieves QS ≈ 0.8 when no information is leaked. This result shows that PRISM* can accurately translate the texts while protecting the privacy of the texts.

Table 2 shows sample translations of PRISM*. The leaked information $x_{\text{pub}}$ does not make sense and reveals little about the secret text $x_{\text{pri}}$. Although it contains some grammatical errors, the output $y_{\text{pri}}$ is generally a correct translation of the input text $x_{\text{pri}}$, which is useful for native speakers to grasp the content.

Table 2: Examples of PRISM* translations. The query $x_{\mathrm{pub}}$ to the translator does not make sense and provides little information on the secret text $x_{\mathrm{pri}}$. Although it contains some grammatical errors, the output $y_{\mathrm{pri}}$ of PRISM* is generally a correct translation of the input text $x_{\mathrm{pri}}$.

| $x_{\mathrm{pri}}$ | $x_{\mathrm{pub}}$ | $y_{\mathrm{pri}}$ |
|---|---|---|
| Jimmy didn't eat breakfast. Because he didn't eat breakfast he was very hungry and could not stop thinking about food. He was thinking about all of the things that he liked to eat the most. He was thinking about breakfast foods like bacon and eggs but he was also thinking about lunch and dinner foods like pizza and macaroni and cheese. | Logan didn't eat hamster. Because he didn't eat circus he was very fishing and could not stop thinking about smile. He was thinking about all of the things that he screamed to eat the most. He was thinking about zoo foods like bacon and grandparents but he was also thinking about guitar and hamburger foods like pizza and lettuce and jungle. | Jimmy n'a pas mangé de déjeuner. Parce qu'il n'a pas mangé de déjeuner, il était très faim et ne pouvait pas arrêter de penser au nourriture. Il pensait à toutes les choses qu'il aimé de vouloir manger le plus. Il pensait aux aliments du déjeuner comme le bacon et les ufs, mais il pensait aussi à des aliments de déjeuner et de dîner comme la pizza et la macaroni et la fromage. |
| A boy was trying to pick out what instrument that he wanted to play. His parents wanted him to pick a good one because playing an instrument was very important to them. So, the boy went to a music store with his parents. | A dragon was trying to pick out what zoo that he wanted to play. His grandchildren wanted herself to pick a good one because playing an Hey was very important to them. Shelly, the bacon went to a mud store with his ants. | Un garçon essayait de choisir quelle instrument il travaillé jouer. Parents parents voulaient eux-mêmes en choisir un bon car jouer avec un Instrument était très important pour eux. So, le garçon est allé dans un magasin de musique avec parents fourmis. |

## 5 Related Work

**Privacy Protection of Texts.** There is a growing demand for privacy protection measures for text data and many methods have been proposed. The U.S. Health Insurance Portability and Accountability Act (HIPAA), which requires that the personal information of patients should be protected, is one of the triggers of heightening concerns on privacy protection of data [5, 20, 29]. One of the challenges to following HIPAA is to protect information hidden in medical records written in free texts [25]. The rule-based method proposed by Neamatullah et al. [28] is one of the early attempts to delete sensitive information from free texts. Li et al. [22] claimed that hiding only the sensitive information is not enough to protect privacy because side information may also leak information and proposed a robust method. Many other methods [12, 23, 26] aim at anonymizing texts so that the authors or the attributions of the authors [39] cannot be inferred. Some methods ensure the rigorous privacy guarantee of differential privacy [7, 43]. The most relevant work to ours is the work by Feyisetan et al. [13], which aims at protecting the privacy of texts while preserving the utility of the texts. Their proposed method is simple enough to implement on the user's side. However, their definition of privacy is different from ours. They aim at protecting the privacy of the author of the text, while we aim at protecting the content. Their method leaks much information on the content of the text. We confirmed this in the experiments. Many of the other methods also aim at protecting the author of the text and keeping the content of the text intact even after the anonymization [7, 12].

**Homomorphic encryption.** Homomorphic encryption [14, 15, 21] enables to compute on encrypted data without decrypting them. The service provider can carry out the computation without knowing the content of the data with this technology [2, 6]. However, users cannot enjoy the benefit of secure computing unless the service provider implements the technology. Homomorphic encryption is notoriously slow [27] and can degrade the performance, and therefore, the service provider may be reluctant to implement it. To the best of our knowledge, no commercial translators use homomorphic encryption. PRISM does not require the service provider to implement it. Rather, PRISM applies homomorphic-like (but much lighter) encryption

on the user's side. PRISM can be seen as a combination of client-side encryption, which has been adopted in cloud storage services [9, 44], and homomorphic encryption.

**User-side Realization.** Users are dissatisfied with services. Since the service is not tailor-made for a user, it is natural for dissatisfaction to arise. However, even if users are dissatisfied, they often do not have the means to resolve their dissatisfaction. The user cannot alter the source code of the service, nor can they force the service to change. In this case, the user has no choice but to remain dissatisfied or quit the service. User-side realization provides a solution to this problem. User-side realization [34, 35] provides a general algorithm to deal with common problems on the user's side. Many user-side algorithms for various problems have been proposed. Consul [37] turns unfair recommender systems into fair ones on the user's side. Tiara [36] realizes a customed search engine the results of which are tailored to the user's preference on the user's side. WebShop [45] enables automated shopping in ordinary e-commerce sites on the user's side by using an agent driven by a large language model. WebArena [46] is a general environment to test agents realizing rich functionalities on the user's side. EasyMark [38] realizes large language models with text watermarks on the user's side. Overall, there are many works on user-side realization, but most of them are on recommender systems and search engines. Our work is the first to protect the privacy of texts on the user's side.

## 6 Conclusion

We proposed a novel problem setting of user-side privacy protection for machine translation systems. We proposed two methods, PRISM-R and PRISM*, to turn external machine translation systems into privacy-preserving ones on the user's side. We showed that PRISM-R is differential private and PRISM* striked a better trade-off between privacy and accuracy. We also proposed an evaluation protocol for user-side privacy protection for machine translation systems, which is valuable for facilitating future research in this area.

**Future Work.** There are several directions for future work. First, the main goal of this paper is to help users grasp the content written in a foreign language without leaking the original content. Although the accuracy is better than those of other privacy protection protocols on the user's side, the translation is far from perfect and there are gramatical errors. Combining the proposed approach with stronger local language models to improve the fluency is an attractive direction for future work. Alternatively, there are many orthogonal approaches to improve the trade-off between privacy and accuracy. For example, Reviewer eSaj suggested to insert some extra (fake) queries to the translator to confuse the attacker. Such approaches can be combined with our approach to further improve the effectiveness.

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

## A Examples

Additional examples of PRISM* translations are shown in Table 3.

Table 3: Examples of PRISM* translations. The query $x_{\text{pub}}$ to the translator does not make sense and provides little information on the secret text $x_{\text{pri}}$. Although it contains some grammatical errors, the output $y_{\text{pri}}$ of PRISM* is generally a correct translation of the input text $x_{\text{pri}}$.

| | |
|---|---|
| $x_{\text{pri}}$ | Oliver is a cat. He has a sister called Spike. Oliver and Spike like to play outside. They chase bugs in the backyard. When they get tired, they sleep in the sun. They don't like to go outside when it is raining. On rainy days Oliver and Spike sit in the window. They watch the rain through the window. Oliver is big and has grey and white fur. His nose is pink. Spike is small and has grey fur. Her nose is the same color as her fur. Spike is round. Oliver is tall. Oliver likes to eat. He worries when there is no food in his bowl. Spike likes to roll in dirt. Sometimes she is smelly. At Christmas time they like to play with the Christmas tree and presents. Oliver climbs the Christmas tree and breaks ornaments. Spike plays with the presents and unwraps them with her claws. |
| $x_{\text{pub}}$ | Jackson is a cat. He has a sister called Logan. Rocksville and Hannah like to play outside. They chase dinosaurs in the backyard. When they get tired, they sleep in the yogurt. They don't like to go outside when it is raining. On fishing days Susan and Josh sit in the jungle. They watch the zoo through the hero. Reggie is big and has grey and seventh castle. His pizza is pink. Martha is small and has grey everywhere. Her baseball is the same hamster as her lion. Jason is round. Vex is tall. Beth likes to eat. He worries when there is no corn in his bowl. Billy likes to roll in king. Linda she is smelly. At Christmas time they like to play with the Christmas tree and criminals. Mandy climbs the Christmas tree and breaks grandparents. Tasha plays with the dragons and unwraps them with her picnic. |
| $y_{\text{pri}}$ | Oliver est un chat. Il a une sœur appelée Spike. Oliver et Spike aiment jouer dehors. Ils poursuivent des bugs dans le jardin. Quand ils sont fatigués, ils dorment dans le soleil. Ils n'aiment pas sortir quand il pleut. Les jours de pluie, Oliver et Spike s'assoient dans la fenêtre. Ils regardent le pluie à travers le fenêtre. Oliver est grand et a un fourrure gris et blanche. Sa nez est rose. Spike est petite et a du gris fourrure. Sa balle de nez est le même couleur que son fourrure. Spike est rond. Oliver est grand. Oliver aime manger. Il s'inquiète quand il n'y a pas de maïs dans son bol. Spike aime se rouler dans le saleté. Parfois, elle sent nourriture. À Noël, ils aiment jouer avec le sapin de Noël et les présentations. Oliver grimpe dans le sapin de Noël et casse les ornements. Spike joue avec les présentations et les déballe avec son griffes. |
| $x_{\text{pri}}$ | One morning, Justin woke up very excited. He was excited because it was his birthday. He went downstairs to eat breakfast.
When he got downstairs his mom said, "Happy Birthday."
"Thank you!" Justin said. "Am I going to have lots of presents?" he asked his mom.
"Yes, Justin. When your friends come over for your birthday party you'll get lots!"
"Hooray!" Justin said, eating his breakfast.
Later that day, Justin's friends came over for his birthday party, and they brought over lots of presents.
They ate cake and ice cream. They played games like tag and played with a football.
After they were done playing Justin got to open his presents. He saw lots of presents. Red ones, blue ones, tall ones, round ones.
"Thank you!" Justin said, as he started to open his presents. He got a basketball, a robot toy, a new bike and some super hero toys!
After all of Justin's friends left, Justin fell asleep very fast because of the exciting day he had and he was happy he got all of those nice things. |
| $x_{\text{pub}}$ | One morning, Josh forgot up very excited. He was excited because it was his birthday. He went downstairs to eat yogurt.
When he got downstairs his mom said, "fishing sauce."
"Thank you?" Martha said. "teach I going to have dinosaurs of vegetables?" he asked his mom.
"Yes, Jason. When your frogs come over for your birthday party you'll get peas."
"Logan." Billy said, eating his garage.
Later that day, Tasha's friends came over for his birthday party, and they brought over criminals of sidewalks.
They slept circus and ice cream. They screamed grandparents like tag and played with a broccoli.
After they were done playing Julia got to open his monkeys. He saw dragons of grandchildren. pink ones, seventh ones, tall ones, round ones.
"Thank you." Jackson said, as he started to open his ants. He got a lettuce, a castle Hey, a new smile and some super jungle toys?
After all of Rocksville's friends left, Susan fell asleep very fast because of the exciting day he had and he was happy he got all of those nice things. |
| $y_{\text{pri}}$ | Un matin, Justin s'est réveillé très excité. Il était excité parce que c'était son anniversaire. Il est descendu en bas pour manger du déjeuner.
Lorsqu'il est arrivé en bas, sa maman a dit : !l'anniversaire happy".
!Merci !" dit Justin. !Est-ce que je vais avoir des lots de présentations !" demanda-t-il à sa maman.
!Oui, Justin. Lorsque tes amis viendront pour ta fête d'anniversaire, tu auras des petits lots".
!Hooray", dit Justin en mangeant son déjeuner.
Plus tard gâteau basket-ball journée, les amis de Justin sont venus pour sa fête d'anniversaire, et ils ont apporté des lots de présentations.
Ils ont mangé comme des circassiens et mangé de basket-ball glace. Ils ont joué jeux comme au jeu de basket-ball !tag" et joué avec un football.
Après avoir fini de jouer, Justin a pu ouvrir ses présentations. Il a vu des lots de présentations. Des rouge, des bleu, des grands, des ronds.
!Merci," dit Justin en commençant à ouvrir ses présentations. Il a eu une salade, un robot Hé, un nouveau bicyclette et des jouets de basket-ball héros super !
Après que tous les amis de Justin soient partis, Justin s'est endormi très vite à cause de cette journée excitante. Il était heureux réveillé reçu amis ces belles choses. |

