# OpenReview forum: "Making Translators Privacy-aware on the User's Side"
_TMLR — Rejected by TMLR_

### Review · Reviewer_eSaj · 2023-12-23

**Summary Of Contributions:**

This paper introduce a method PRISM with two variants, PRISM-R and PRISM*, to deal with the privacy for translation service. The basic idea involves randomly replace words from the original input texts, and send randomized texts to the translation services to get a translated results back. Then the randomized words in the targeted language will be changed to the original words in the translated language. The paper showed theoretically they can show the DP between the original texts and randomized texts.

**Audience:**

Yes

**Broader Impact Concerns:**

The problem is very important and timely, and can produce positive impacts to the society and public.

**Claims And Evidence:**

Yes

**Requested Changes:**

1. Given the input (original) texts and randomized texts, how to evaluate the "privacy leakage" between them? DP used in the paper is not appropriate.
2. The method likely need to consider how much randomization can be done for replacement. A suggestion is to consider adding some extra sentences so that the "attacker" may not know what the user really want to translate.
3. Providing some more detailed examples in the appendix to show how the approaches actually work.

**Strengths And Weaknesses:**

Strengths:
+ Due to the fast adoption of generative AI technology and services, such as GPT, Bard, Google/Translation, privacy issues for generative AI becomes increasingly critical. The research problem is both timely and important
+ The solution has some merits
+ The paper is easy to read

Weaknesses:
- The privacy issue of translation is not properly addresses and the proposed solution may not be sufficient for this problem
- The differential privacy used here is inappropriate due to the dependency between words.
- More experimental results and evaluation criteria should be designed to evaluate privacy for this task

---

> ### Author Response · Authors · 2024-01-27
>
> Thank you for the comments.
>
> > Given the input (original) texts and randomized texts, how to evaluate the "privacy leakage" between them?
>
> This is an essentially difficult problem. There have been many attempts to evaluate the ``intrinsic difference'' between texts, such as Word Mover's Distance [Kusner+ ICML 2015], BERTScore [Zhang+ ICLR 2020], and G-EVAL [Liu+ EMNLP 2023]. However, none of them has no drawbacks, see, e.g., [Fabbri+ TACL 2021]. Proposing a good metric for this approach is beyond the scope of this paper. Therefore, we sidestepped directly evaluating "privacy leakage" between two texts and proposed evaluating "privacy leakage" by utilizing auxiliary question answering datasets and a privileged information approach, see Section 4.1.
>
> - [Kusner+ ICML 2015] From Word Embeddings To Document Distances.
> - [Zhang+ ICLR 2020] BERTScore: Evaluating Text Generation with BERT.
> - [Liu+ EMNLP 2023] G-Eval: NLG Evaluation using GPT-4 with Better Human Alignment.
> - [Fabbri+ TACL 2021] SummEval: Re-evaluating Summarization Evaluation.
>
>
> > The method likely need to consider how much randomization can be done for replacement. A suggestion is to consider adding some extra sentences so that the "attacker" may not know what the user really want to translate.
>
> Thank you for suggesting an interesting approach! As this approach is orthogonal to our method, it can be combined with PRISM and further strengthens privacy. We have added some discussions on this approach in the Future work section.
>
> > Providing some more detailed examples in the appendix to show how the approaches actually work.
>
> We have added some more examples in the Appendix.

---

### Review · Reviewer_23Xs · 2023-12-28

**Summary Of Contributions:**

This manuscript introduces PRISM-R and PRISM*, which are algorithms designed to modify the input and output of translation models to ensure that input query privacy is respected.

The core idea is to switch out parts of the input phrase with similar types of words that do not dramatically change the translation, and then post translation, simply replace those initial words with the direct word-for-word translation of the original word. This way, none of the original messages are sent directly to the platform.

To evaluate the algorithm the authors develop a privacy score and a quality score, using a question-answer dataset. For the privacy score, an external evaluator (gpt-3.5) answers the question using the input to the translator. For the quality score, the platform uses the final output and asks the evaluator to use that to evaluate the output.

**Audience:**

Yes

**Broader Impact Concerns:**

Are there any systematic biases specific to prism? Can any studies related to this be conducted?

**Claims And Evidence:**

Yes

**Requested Changes:**

"PRISM effectively balances privacy protection with translation accuracy"

Mostly, I would like to see a more nuanced treatment of the above statement, based on my comments above.

**Strengths And Weaknesses:**

Weaknesses:
1. I don’t think this algorithm is a good idea (see three basic failure modes I highlighted). It seems like a good place to start, but the authors need to develop a more sophisticated solution that deals with at least some of the issues I discuss below.

2.  Even if it leads to an information-complete result, I’m not sure the end result is actually better than what you could get from an on-device translation software.

3. The first example of table 2 is awesome in the sense that there are some very interesting issues which arise due to the specific substitutions. I would like to see this discussed in detail.

* Breakfast (which is one of the swapped words) is mis-translated. Is this because “breakfast” is translated to “petit-dejeuner” (small-lunch) which is a multi-word translation? Google translate has no trouble with this.

* Another interesting example is the substitution “liked”->”screamed”. In this case “liked” is meant to be used in the “present imperfect” way, not in a passed tense way. “Screamed” would never be used in this way. PRISM’s translation is aim’/e, which is a past tense, and is a poor translation. Again google translate has no issues with this, and uses the imperfect “amait”.

* In French, I don’t think anyone would say “la macaroni et la fromage”, more likely they would say “les macaronis au fromage”. This translation is probably a side effect of treating mac+cheese as two separate nouns and swapping them, as opposed to treating it like a single entity.

---

> ### Author Response · Authors · 2024-01-27
>
> Thank you for the comments.
>
> > Even if it leads to an information-complete result, I’m not sure the end result is actually better than what you could get from an on-device translation software.
>
> Our approach is extremely lightweight. The only requirement is the word dictionary, and no language models are required on the device. Therefore, the strengths of our approach and on-device translation software complement one another. Devices with no network access may prefer on-device translation software while devices with low computational and storage resources may prefer our approach.
>
> > The first example of table 2 is awesome in the sense that there are some very interesting issues which arise due to the specific substitutions. I would like to see this discussed in detail.
>
> As we are not native speakers of French, we cannot discuss these grammatical errors in detail.
> However, we are aware of grammatical errors. We noted `Although it contains some grammatical errors, the output y_pri of PRISM* is generally a correct translation of the input text x_pri.` in the caption of Table 2, and we noted `Although it contains some grammatical errors, the output y_pri is generally a correct translation of the input text x_pri, which is useful for native speakers to grasp the content.` in the main text.
> We emphasize that our goal is not to provide perfect translations, but to enable native speakers to grasp the content.
>
> Suppose the input text x_pri is the following Japanese text.
> ```
> ジミーは朝食を食べなかった。朝食を食べなかったため、彼はとても空腹で、食べ物のことが頭から離れなかった。彼は自分が一番好きな食べ物のことばかり考えていた。ベーコンエッグのような朝食の食べ物だけでなく、ピザやマカロニチーズのような昼食や夕食の食べ物についても考えていた。
> ```
>
> You may not understand the contents at all. As this text is secret, you cannot use Google Translate. PRISM provides the following translation:
> ```
> Jimmy n’a pas mangé de déjeuner. Parce qu’il n’a pas mangé de déjeuner, il était très faim et ne pouvait pas arrêter de penser au nourriture. Il pensait à toutes les choses qu’il aimé de vouloir manger le plus. Il pensait aux aliments du déjeuner comme le bacon et les ufs, mais il pensait aussi à des aliments de déjeuner et de dîner comme la pizza et la macaroni et la fromage.
> ```
>
> Indeed, this is not a perfect translation, but you could grasp what the original Japanese tries to tell. This is what PRISM aims at.
>
> > "PRISM effectively balances privacy protection with translation accuracy"
> > Mostly, I would like to see a more nuanced treatment of the above statement, based on my comments above.
>
> We agree that we could not tell the goal and the use cases well in the original manuscript. We will state our claims more carefully in the camera ready and emphasize (i) the confirmed effectiveness is relative to other user-side privacy preservation protocols and (ii) our goal is not to provide perfect translations, but to enable native speakers to grasp the content written in a foreign language.

---

> > ### Comment · Reviewer_23Xs · 2024-02-09
> > **Response to Comment**
> >
> > I have read your comments. I agree that the text is intelligible, but I do not think it is okay for you to say "we are not native speakers of French, we cannot discuss these grammatical errors in detail". These errors constitute failures of your algorithm--and exemplify its many modes of failure.
> >
> > I think we must hold researchers to a higher standard than this.
> >
> > 1. As you have chosen to include this example in your paper, you should study this example and characterize these errors.
> >
> > 2. On-device translation tools have been available for a long time now, and I think they perform reasonably well. Though you don't have to be SOTA with respect to them, I would like to see a comparison. See
> > https://developers.google.com/ml-kit/language/translation.
> >
> > Please discuss the computational cost of these approaches as compared to yours in real-world settings.

---

> > > ### Author Response · Authors · 2024-02-09
> > >
> > > Thank you for your response.
> > >
> > > > As you have chosen to include this example in your paper, you should study this example and characterize these errors.
> > >
> > > We try our best to discuss these examples. Please correct us if we made mistakes. We would really appreciate it if you could help us polish the discussions.
> > >
> > > > Breakfast (which is one of the swapped words) is mis-translated. Is this because “breakfast” is translated to “petit-dejeuner” (small-lunch) which is a multi-word translation?
> > >
> > > Yes. More concretely, we found this was caused by the word dictionary we created in an unsupervised way. We computed probability ratio (Eq. 14)  tokenwise, and as a result, the first candidate was L("breakfast", "NN", 1) = "déjeuner" and the second candidate was L("breakfast", "NN", 2) = "petit". Thus, this problem is resolved by using a more sophisticated dictionary with L("breakfast", "NN", 1) = "petit déjeuner" such as a manucally compiled one.
> > >
> > > > Another interesting example is the substitution “liked”->”screamed”. In this case “liked” is meant to be used in the “present imperfect” way, not in a passed tense way. “Screamed” would never be used in this way. PRISM’s translation is aim’/e, which is a past tense, and is a poor translation.
> > >
> > > As far as we understand, "liked" is in a past tense used for backshift of tenses. We hypothesize that the mistranslation is caused because there are no one-to-one correspondence between the tenses in English and French. This is one of the limitations as PRISM implicitly assumes tenses basically corresnpond.
> > >
> > > > In French, I don’t think anyone would say “la macaroni et la fromage”, more likely they would say “les macaronis au fromage”. This translation is probably a side effect of treating mac+cheese as two separate nouns and swapping them, as opposed to treating it like a single entity.
> > >
> > > We agree with your point. This mistranslation is caused by treating macaroni and cheese separately. This could be resolved by using N-gram translation dictionaries instead of tokenwise dictionary but it requires more computational and storage cost. We prefer to keep it in its present simple form because we consider that PRISM already servs its purpose of conveying rough meaning for non-native speakers, and that simplicity and lightweight are important to broaden the usage.

---

> > > > ### Author Response · Authors · 2024-02-09
> > > >
> > > > > On-device translation tools have been available for a long time now, and I think they perform reasonably well. Though you don't have to be SOTA with respect to them, I would like to see a comparison.
> > > >
> > > > The advantages of our approach over Google ML Kit's on-device translation API are (i) Google ML Kit consumes about 30MB of stroage per language and 1.7GB for all (of 58) languages, which could be prohibitive for edge devices. PRISM consumes only abount 2.2MB of storage per language and 128 MB for 58 languages, which are more than 10 times more storage efficient. (ii) Google ML Kit has usage restrictions `ML Kit’s on-device Translation API may not be used in any applications for any embedded devices such as cars, TVs, appliances, or speakers without Google's prior written permission. This Service can only be integrated with applications for the following personal computing devices: smartphones, tablets, and laptop and desktop computers.` https://developers.google.com/ml-kit/language/translation/translation-terms while PRISM can be combined with any translators including open ones.
> > > >
> > > > An apples-to-apples comparison is difficult for PRISM and Google ML Kit because Google ML Kit can be used only from Java, Cotlin, Swift, or Objective-C while PRISM is implemented by Phthon, and it is difficult to control the experimental conditions. Thus we conducted experiments using
> > > > TinyLlama, which is one of the state-of-the-art lightweight LLMs. The following table summarizes costs of both approaches.
> > > >
> > > > |     | Storage Consumption | Time Consumption on device |
> > > > | --- | --- | --- |
> > > > | PRISM on CPU |  2.2MB   |  0.09s   |
> > > > | TinyLlama on CPU | 2.2GB | 17.65s |
> > > > | TinyLlama on RTX 3060 GPU | 2.2GB | 1.52s |
> > > >
> > > > PRISM is 1000 times more storage efficient, and 196 times more time efficient. The translation results are as follows:
> > > >
> > > >
> > > > Input
> > > > ```
> > > > Jimmy didn’t eat breakfast. Because he didn’t eat breakfast he was very hungry and could not stop thinking about food. He was thinking about all of the things that he liked to eat the most. He was thinking about breakfast foods like bacon and eggs but he was also thinking about lunch and dinner foods like pizza and macaroni and cheese.
> > > > ```
> > > >
> > > > Output of PRISM
> > > > ```
> > > > Jimmy n’a pas mangé de déjeuner. Parce qu’il n’a pas mangé de déjeuner, il était très faim et ne pouvait pas arrêter de penser au nourriture. Il pensait à toutes les choses qu’il aimé de vouloir manger le plus. Il pensait aux aliments du déjeuner comme le bacon et les ufs, mais il pensait aussi à des aliments de déjeuner et de dîner comme la pizza et la macaroni et la fromage.
> > > > ```
> > > >
> > > > Output of TinyLlama
> > > > ```
> > > > Jimmy ne pas manger le matin. C'est-à-dire qu'il n'a pas mangé le matin il était très surpeuphoné et ne pouvait pas arrêter de penser aux aliments qu'il apprécie le plus. Il était pensant sur les choses que il apprécie le plus. Il était pensant sur les aliments de l'appétit comme le pain et l'oeuf, mais il était également pensant sur les aliments de dîner comme le pizza et le fromage gratiné.
> > > > ```
> > > >
> > > > Athough TinyLlama is fluent, we found the translation quality were not much better than that of PRISM. TinyLlama did not translate "hungry", "breakfast foods", and "lunch" well. These results indicate that PRISM is more lightweight than on-device translators with comparable translation quality.

---

### Review · Reviewer_kmqQ · 2024-01-24

**Summary Of Contributions:**

This paper introduces the "PRISM" method, allowing users of machine translation systems to safeguard data privacy. The proposed method is evaluated through experiments conducted on T5 and ChatGPT, focusing on English → French and English → German translations.

**Audience:**

Yes

**Claims And Evidence:**

Yes

**Requested Changes:**

See above waknesses.

**Strengths And Weaknesses:**

Strengths:
1. The introduced "PRISM" method empowers users to safeguard data privacy, shifting the responsibility from the server.
2. The authors present two variants of "PRISM". "PRISM-R" has a theoretical guarantee of differential privacy, and "PRISM* (PRISM-Star)" is a more advanced method that achieves superior translation accuracy compared to PRISM-R, albeit at the expense of losing the theoretical guarantee.

Weaknesses:
1. The study could benefit from exploring additional datasets and translation tasks, such as French → English.
2. While "PRISM" exhibits higher translation accuracy than other privacy-preserving translation technologies, it would be valuable to understand the accuracy loss compared to public translation.
3. The impact of token length on the effectiveness of the proposed "PRISM" is not fully explored.

---

> ### Author Response · Authors · 2024-01-27
>
> Thank you for the comments.
>
> > While "PRISM" exhibits higher translation accuracy than other privacy-preserving translation technologies, it would be valuable to understand the accuracy loss compared to public translation.
>
> In Figure 3, The top-left point of each line indicates the score of the public translation (without any privacy preserving measures). The drop in y-axis indicates the quality loss compared to public translation, and the x-axis indicates the privacy benefit. These results show that PRISM* is not much worse than public translation (and PRISM* provides privacy protection).
>
> > The impact of token length on the effectiveness of the proposed "PRISM" is not fully explored.
>
> We have conducted an additional experiment. We split the dataset based on the token length and measured the AUPQC of PRISM* for each split.
>
> | #words | ~800  | 800 ~ 1000 | 1000 ~ 1200 | 1200 ~ 1400 | 1400 ~ |
> | --- | ----- | ---------- | ----------- | ----------- | ------ |
> | AUPQC | 0.505 | 0.412      | 0.551       | 0.496       | 0.520  |
>
> These results indicate the token length does not affect the effectiveness much. Note that the results have high variance due to the dataset size and data splitting, but as the baseline methods has < 0.4 AUPQC (see Table 1), these results show that PRISM* outperforms the baseline methods in all levels.

---

### Author Response · Authors · 2024-03-20

Dear Reviewers and AE,

Thank you for handling our paper.
We answered the reviewers' questions on 27 Jan and 09 Feb. We suppose we could resolve most of the concerns.
If you have any remaining concerns, please feel free to let us know. We are happy to discuss them.

Best Regards

---

### Decision · Action_Editor_YQaM · 2024-03-20

**Recommendation:** Reject

**Comment:**

The evidence provided does not support the claims made. Please see the "Claims And Evidence" box for details.

**Audience:**

Yes, this is of interest to at least some subset of TMLR's audience

**Claims And Evidence:**

The paper's main claim is to 'protect [with respect to privacy] data on the user's side'. The paper contrasts this claim with prior work that in the latter 'sophisticated adversaries might still extract sensitive information'. However, the reviewers make several criticisms of the claims made in the paper. The primary issue pointed out, which belies the claims, is as follows. When defining the notion of (differential) privacy, the paper considers two databases as adjacent if the two texts differ in one word.  This is an issue because the words in the text of a user are all correlated, and a certain user corresponds to a block of text rather than a word. Here, individual words don't correspond to user records, so protecting privacy with respect to individual words does not correspond to protecting privacy of individual users. The actual privacy provided does not match the claims made about it.

---

> ### Author Response · Authors · 2024-03-23
>
> We respectfully disagree that the privacy definition has a crucial issue. Even when multiple words are changed, we can reach one state from the other state by hopping the adjacency relations. Thus, the same guarantee of differential privacy can be derived immediately where two texts are regarded adjacent if an entire block in the text is changed or if a constant number of words are changed.
>
> We wish we could have discussed this with you before the decision.